# Metabolite dynamics over the course of anti-tuberculosis treatment in individuals with mild and severe tuberculosis

Catherine H Kagemann[1,2], Senbagavalli Prakash Babu[3]*, Komala Ezhumalai[3], Arnab Chakraborty[4], Kalaivani Raghupathy[3], Siddhesh S Kamat[4], Vijay Viswanathan[5], Samantha L Huey[1,2], Prakash Babu Narasimhan[1,3,6], Pranay Sinha[7,8], Elaine A Yu[9], Saurabh Mehta[1,2,10]*, Sonali Sarkar[3], on behalf of the Regional Prospective Observational Research on Tuberculosis India Consortium[¶]

1 Cornell Joan Klein Jacobs Center for Precision Nutrition and Health, Cornell University, Ithaca, New York, United States of America, 2 Division of Nutritional Sciences, Cornell University, Ithaca, New York, United States of America, 3 Department of Preventive and Social Medicine, Jawaharlal Institute of Postgraduate Medical Education and Research, JIPMER Campus Rd, Gorimedu, Dhanvantari Nagar, Puducherry, India, 4 Department of Biology, Indian Institute of Science Education and Research (IISER), Pashan, Pune, Maharashtra, India, 5 Department of Diabetology, Prof M Viswanathan Diabetes Research Centre (MVDRC), Chennai, India, 6 Mahatma Gandhi Medical Advanced Research Institute (MGMARI), Sri Balaji Vidyapeeth, SBV Campus, Central Office of Research, Pillaiyarkuppam, Puducherry, India, 7 Boston Medical Center, Section of Infectious Diseases, Boston, Massachussets, United States of America, 8 Boston University Chobanian Avedisian School of Medicine, Section of Infectious Diseases, Boston, Massachussets, United States of America, 9 Vitalant Research Institute, San Francisco, California, United States of America, 10 Division of Medical Informatics, St. John's Research Institute, Bengaluru, Karnataka, India

¶ Membership of the Tuberculosis India Consortium is listed in the Acknowledgments.
* smehta@cornell.edu (SM); prco.indoustb@gmail.com (SPB)

## Abstract

Mycobacterium tuberculosis (Mtb) manipulates host metabolism to gain nutrients and increase virulence. Despite known alterations in metabolism in individuals with pulmonary tuberculosis (PTB) during anti-tuberculosis (TB) treatment, the effect of disease severity on metabolite dynamics in individuals with PTB remains understudied. We examined metabolite dynamics over the course of anti-TB treatment in individuals diagnosed with mild (N = 8; smear grade of 1 + /2+ and mild chest x-ray (CXR) abnormality) or severe drug-sensitive PTB (N = 8; 3 + smear grade and moderate/advanced CXR abnormality) in a pilot proof-of-concept study compared to controls without TB (N = 7). Semi-targeted metabolomic analysis of plasma was performed using tandem liquid chromatography-mass spectrometry and electrospray ionization mass spectrometry at baseline, one month, and six months after treatment initiation. Our analysis revealed disease severity-specific metabolic profiles as well as those unique to controls. Many metabolites specific to mild or severe TB were involved in the glycerophospholipid and sphingolipid pathways. A subset of glycerophospholipids were enriched at baseline, month 1, and at the endpoint in individuals with mild and severe TB, despite anti-TB treatment. Our results highlight the importance of glycerophospholipid and sphingolipid pathways during Mtb infection and treatment,

**Data availability statement:** All the data and materials supporting this manuscript are within the paper and the supplementary file, also accessed at the Zenodo repository (https://doi.org/10.5281/zenodo.16944363).

**Funding:** This project was funded in part by the Government of India's Department of Biotechnology, the United States National Institutes of Health, and the National Institute of Allergy and Infectious Diseases, with distribution by CRDF Global, the National Academy of Sciences and the United States Agency for International Development under the Prime Award Number AID-OAA-A-11-00012 (PEER, SPB). Authors SS and SPB received funding from the Regional Prospective Observational Research for Tuberculosis India Consortium (RePORT). Research reported in this publication was also supported by the Office of the Director and Eunice Kennedy Shriver National Institute of Child Health & Human Development of the National Institutes of Health under award 1 T32 HD113301 (CHK). The content is solely the responsibility of the authors and does not necessarily represent the official views of the National Institutes of Health. The funders had no role in study design, data collection and analysis, decision to publish, or preparation of the manuscript. Mention of trade names or organizations does not imply endorsement by any sponsors.

**Competing interests:** The authors have declared that no competing interests exist.

regardless of disease severity, and suggest that Mtb could induce chronic effects on host metabolism even after treatment.

## Introduction

*Mycobacterium tuberculosis* (Mtb), the causative agent of tuberculosis (TB), continues to pose a significant global health threat despite decades of efforts towards eradication [1–3]. The World Health Organization (WHO) recently revealed that 8.2 million people were diagnosed with TB in 2023 making TB resurface as the leading infectious disease globally [3].

Central to the resilience of Mtb is its ability to manipulate host metabolites, enhancing its virulence and ensuring its survival within the host [4–8]. Metabolomics has been used to advance therapeutics and biomarker detection for diseases such as malaria and HIV [9–13]. However, the application of metabolomics to TB research is still in its early stages. For example, there is a lack of comprehensive metabolomics studies across various stages of disease severity and treatment. While numerous omics studies have been conducted to test the effect of disease severity, treatment timepoints, and sex-specific responses, a notable gap exists in the realm of TB metabolomics concerning these variables [14–19].

It is known that Mtb infection and the amount of anti-TB treatment received can influence host metabolism [4,6,7,8,20–25]. However, to our knowledge, it is unclear whether host metabolism is affected by the severity of active TB disease. Two studies investigating whether host metabolic profiles differ between mild and severe tuberculosis (TB) reported no significant differences [26,27]. However, severity classification in these studies relied solely on chest X-ray (CXR) lesions, which may not fully capture disease activity [26,27]. Importantly, our analysis incorporates sputum smear grading—a key indicator of mycobacterial burden. While CXR abnormalities reflect structural lung damage, they do not always correlate with bacterial activity; for instance, extensive lesions can be present even in less active or early-stage disease. Including both CXR and sputum smear profiling provides a more comprehensive assessment of disease severity and is crucial for accurately evaluating the host's metabolic response to TB. Therefore, gaining a comprehensive understanding of how TB influences host metabolism over the course of anti-TB treatment and in individuals with different disease severities is imperative for unraveling the complexities of TB pathogenesis and identifying potential future therapeutic targets.

In this study, we analyzed metabolite dynamics throughout the course of anti-TB treatment, from baseline to month 6, in individuals with mild or severe pulmonary TB (PTB) in India. The study advances our understanding of TB pathogenesis by unraveling the intricate metabolic alterations associated with disease severity and treatment response.

## Materials and methods

### Ethics

The study protocol was approved by the Scientific Advisory Committee and Ethics Committee of Jawaharlal Institute of Postgraduate Medical Education and

Research. Permission to utilize the samples and data of RePORT India Common Protocol (CP) was sought from the RePORT India Executive Committee (EC). We used data collected Jawaharlal Institute of Postgraduate Medical Education and Research and Prof. M. Viswanathan Diabetes Research between 2017 and 2019, which are part of the RePORT India Consortium [28].

## Study design: Prospective cohort analysis

We analyzed data from longitudinal observational studies among individuals with PTB across the Clinical Research Sites (CRS) in the southern and western parts of India under Regional Prospective Observational Research for Tuberculosis (RePORT)-India consortium [28,29]. This study was designed as a pilot investigation utilizing retrospectively available samples, with the primary aim of exploring metabolic differences across varying severities of PTB.

The RePORT India CP Phase I study involved the collection of pre-determined clinical data and biological specimens at specified treatment timepoints, using a unified protocol and standardized methods, and aimed to provide specimens to biomarker researchers to facilitate TB research. Samples and data were collected after obtaining informed consent from adults (≥18 years). The recruitment period was from 8th September 2017 – 9th January 2019. For the present study, inclusion criteria of individuals were age ≥ 18 years, confirmed PTB as determined by a positive acid-fast bacilli (AFB) sputum smear, and confirmed by a positive sputum culture for Mtb, ≤ 5 days of anti-TB treatment in their lifetime and a Karnofsky score <40 (N = 187). Exclusion criteria included female sex, unknown/positive human immunodeficiency virus (HIV) status, diagnosis of or contact with someone with known drug-resistant TB, diagnosis of diabetes mellitus (types 1 and 2), cancer, chronic kidney failure, an active psychiatric condition, or alcohol or drug dependence (N = 171). Participants of the female sex were excluded along with comorbidities (diabetes, smoking, alcohol, HIV) to focus on understanding TB pathogenesis. Biological sex-based metabolic differences and comorbidities could confound results. Control inclusion criteria included individuals whose household contacts were age ≥ 18 years, significant contact with PTB cases for at least 5 days/week or bedroom sharing, and interferon-gamma release assay negative (N = 191). Exclusion criteria for controls included female sex, unknown/positive HIV status, history of diabetes mellitus and symptoms suggestive of TB in the last 3 months such as cough, fever, hemostasis, weight loss, chest pain, fatigue, night sweats and loss of appetite (N = 184).

Eligible participants in the study were categorized into three groups based on clinical and radiological severity of PTB and infection status. CXR scoring was performed with a total possible score of 140. The lungs were divided into six zones, and each zone was assessed for abnormalities such as scarring and cavitation. Scores were assigned based on the extent and nature of these findings in each zone. *Group 1* (Severe group; N = 8) comprised individuals with moderate to advanced CXR abnormalities and 3 + sputum smear positivity. *Group 2* (Mild group; N = 8) included participants with mild CXR abnormalities and 2 + / 1 + sputum smear positivity. *Group 3* (Control group; N = 7) consisted of IGRA-negative, healthy household contacts of PTB patients from Groups 1 and 2, who followed similar dietary patterns and served as metabolic controls.

All individuals with PTB were drug-susceptible and followed the same DOTS (Directly Observed Treatment Short Course) treatment regimen, which included two months of intensive phase treatment and 4 months of continuation phase treatment. The intensive phase treatment consisted of 8 weeks of isoniazid, rifampicin, pyrazinamide and ethambutol in daily doses as per four weight band categories. After two months, pyrazinamide was stopped and rest of the three drugs were continued in the continuation phase for another 16 weeks as daily dosages.

Peripheral blood samples were obtained by venipuncture from all the participants during their respective study visits. Whole blood collected in sodium heparin tubes was centrifuged and the isolated plasma was immediately stored at -80 °C at the clinical site of CP Phase I. Samples were shipped on dry ice from the clinical site to the central biorepository at National Institute of Research in Tuberculosis (NIRT), Chennai. Height (cm) and weight (kg) were measured, and height was rounded if the measurement was equal to or greater than 0.5. Body mass index

(BMI, exact m/kg$^2$) was measured and categorized as underweight (<18.5 kg/m2), normal (18.5–22.9 kg/m2), over-weight (23–24.9 kg/m2) and obese (≥25 kg/m2) based on the Asian classification [30]. Following approval from the EC of RePORT India, the plasma samples were shipped from the biorepository to JIPMER for metabolite extraction. Metabolomic analysis was performed at Indian Institutes of Science Education and Research (IISER), Pune. Associated sociodemographic and clinical data collected were requested from the data coordination centre, RePORT India. The data were accessed for research purposes on Oct 18, 2023. No retrospective data was received without access to identifiable information.

Metabolomic analysis of the plasma samples was performed for Group 1, N = 8 severe PTB and Group 2, N = 8 mild PTB at baseline, one month after treatment initiation (month 1) and at the end of treatment (EOT) at month 6; for Group 3, controls (N = 7) only baseline samples were analyzed.

## Metabolomic analysis

### Liquid chromatography and mass spectrometry analysis for lipids

Plasma samples (0.5 mL) were prepared by adding a 2:1:1 mixture of chloroform (CHCl$_3$), methanol (MeOH), and plasma to a 2 mL final volume, containing 1 nmol of 17:0–20:4 PC (Avanti Polar Lipids) as internal standards for a semi-quantitative analysis. Blanks with resuspension solvents were run before each group of samples (control, mild, severe TB). This homogenate mixture was vigorously vortexed and centrifuged at 2800g for 10 minutes to separate the mixture into an organic phase (bottom) and an aqueous phase (top) separated by a protein disk. The organic phase was removed by pipetting and stored on ice (at 0–4 °C), while 50 µL of formic acid was added to the aqueous phase to enhance the extraction of phospholipids. This mixture was vigorously vortexed and 1 mL of CHCl$_3$ was added, after which the mixture was again mixed. Post centrifugation was conducted using the same conditions and the organic layer from this extraction was pooled with the previously obtained organic layer and dried under a stream of nitrogen gas. The dried lipid extracts were re-solubilized in 200 µL of 2:1 CHCl$_3$: MeOH and 10 µL was injected into an Agilent 6545 LC-QTOF (quadrupole-time-of-flight) LC-MS/MS for high-resolution auto MS-MS methods and chromatography techniques. A Gemini 5U C-18 column (Phenomenex) coupled with a Gemini guard column (Phenomenex, 4x3 mm, Phenomenex security cartridge) was used for LC separation. The solvents used for negative ion mode were: buffer A: 95:5 H$_2$O: MeOH + 0.1% ammonium hydroxide and buffer B: 60:35:5 iPrOH: MeOH: H$_2$O + 0.1% ammonium hydroxide. The 0.1% ammonium hydroxide in each buffer was replaced by 0.1% Formic acid + 10mM ammonium formate for positive ion mode runs. We started LC separation with 0.3 mL/min 100% buffer A for 5 minutes, 0.5 mL/min linear gradient to 100% buffer B over 40 minutes, 0.5 mL/min 100% buffer B for 10 minutes, and equilibration with 0.5 mL/min 100% buffer A for 5 minutes.

The following settings were used for the ESI-MS analysis: drying gas and sheath gas temperature: 320 °C, drying gas and sheath gas flow rate: 10L/min, fragmentor voltage: 150V, capillary voltage: 4000V, nebulizer (ion source gas) pressure: 45 psi and nozzle voltage: 1000V. For analysis, a lipid library curated from available literature regarding lipids and metabolites previously reported in mammalian systems [6,8,20,31,32], was employed in the form of a Personal Compound Database Library (PCDL), and the peaks were validated based on relative retention times and fragments obtained (S1 Table). The PCDL was generated from Metlin, Lipid Maps and Lipid Maps CompDB over many years based on detection across different extracts by using MS1 and MS2 signatures (no isotope standards). Lipid species were quantified by normalizing their areas under the curve (AUC) to the internal standard added. Our database contains all major groups of glycerophospholipids, sphingolipids, and neutral lipids with individual tails ranging from 14 to 24 chains and different degrees of unsaturation. In total, we used 165 of 261 lipids from our PCDL as some did not pass our stringency criteria of detection. The following stringency criteria for detection were used: an MS1 error cutoff of 10 ppm, a minimum EIC peak height of 1000, and ion detection in positive mode for [M + H]+ and [M + NH4]+ ions, as well as in negative mode for [M-H]- ions.

## Computational analyses

### Differential abundance analysis

We utilized linear regression models followed by a differential abundance analyses in limma [33] to assess changes in individual metabolites in individuals infected with TB across disease severities and treatment stages. No imputation was conducted on our data. The AUC of the metabolites were log transformed as the majority of the metabolites fit under a log normal distribution prior to log transformation. We used the following linear regression models: 1) Y (AUC of metabolites) = $\beta_0 + \beta_1 \times$ Disease Severity, 2) $Y = \beta_0 + \beta_1 \times$ Treatment Time, and 3) $Y = \beta_0 + \beta_1 \times$ Disease Severity $+ \beta_2 \times$ Treatment Timepoint $+ \beta_3 \times$ (Disease Severity $\times$ Treatment Time). Following fitting our data to the three linear regression models, we conducted differential abundance analyses to determine which metabolites had significant increases or decreases in abundance compared to the controls ($p < 0.05$, Benjamini-Hochberg FDR correction, absolute log2 fold change $> 0.5$). We specifically focused on the differences between treatment and the control, as between-treatment comparisons may have included metabolites that were not differentially abundant in the control. Heatmaps and volcano plots were made using ggplot2 in R (version 4.1.0).

### LIPEA lipid pathway enrichment

We conducted a lipid enrichment analysis using Lipid Pathway Enrichment Analysis (LIPEA) of the differentially abundant lipid metabolites identified from limma [33]. LIPEA uses the Kyoto Encyclopedia of Genes and Genomes (KEGG) database [34–36] and conducts an over representation analysis for each pathway.

### MetaboAnalyst non-lipid pathway enrichment

Metabolite set enrichment analysis (MSEA) [37] was conducted on the differentially abundant non-lipid metabolites identified from our limma analysis using Metaboanalyst [38]. Metabolite sets are comprised of metabolites identified by A Reference list of Metabolite names (RefMet) [39,40], LIPID Metabolites and Pathways Strategy (LIPID MAPS), and KEGG pathways [40].

### Lipid metabolite pathway/interaction networks

Lipid pathway networks were made using LIPID MAPS Bioinformatics Methodology For Pathway Analysis (BioPAN) [40]. BioPAN was utilized on the log transformed area under the curve metabolite data between individuals with severe or mild TB with the controls at each treatment timepoint. BioPAN calculates a z-score to determine whether a given pathway is significantly different between the control and treatment [41]. However, the z-score analysis in BioPAN cannot capture interaction effects and is not suitable for data with large variance. Therefore, we used BioPAN only to determine how the metabolites interact with one another and what pathways they belong to, and subsequently remade the networks in BioRender [41] to show if the metabolite had increased or decreased abundance compared to the healthy controls from our linear regression model ($Y = \beta_0 + \beta_1 \times$ Disease Severity).

## Results

### Baseline characteristics by treatment

Baseline characteristics of the study participants are summarized in S2 Table. The majority of individuals with severe TB were between 40–60 years of age, and 87.5% of all individuals were underweight. Controls were evenly distributed across age groups; the majority were overweight or obese. All individuals with severe TB exhibited potential financial burdens, as reflected by the percentage earning between 3000–10000 Indian rupees (INR) per month. A majority of individuals with severe TB reported current smoking (87.5%) and alcohol use (87.5%). Weight loss within the last 4 weeks prior to baseline data collection (self-reported) was observed in both individuals with mild (100%) and severe (75%) TB. The sputum conversion rate was 75% in individuals with severe TB and 62.5% in individuals with mild TB.

## Mtb-specific enrichment of glycerophospholipid and sphingolipid pathway metabolites

A linear regression model was used to determine the impact of the severity of TB on the area under the curve (AUC) of metabolites at each treatment timepoint. Differential abundance of statistically significant metabolites was subsequently calculated for individuals with mild or severe TB compared to the controls (Fig 1). Of the differentially abundant metabolites in individuals with mild TB as compared to the control, 5 of 21 were non-lipid metabolites, while the remaining 16 were lipids. Similarly, 5 of 29 of the differentially abundant metabolites were non-lipid metabolites while the remaining 24 were lipids in individuals with severe TB as compared to the control. Increased glycerophospholipid abundance was observed at each subsequent treatment timepoint in both mild and severe TB (with the exception of LPC 20:0 and LPE 22:1) compared to the control. These glycerophospholipids included lysophosphatidylinositol (LysoPI) 18:1, LysoPI 18:2, phosphatidylethanolamine (PE) 36:1, and phosphatidylinositol (PI) 36:4.

A total of 165 differentially abundant lipids from individuals with mild or severe TB compared to the controls at each treatment timepoint were combined for lipid pathway enrichment analysis. The analysis showed glycerophospholipid metabolism and sphingolipid signaling pathway enrichment at all timepoints (Fig 2). The low number of differentially abundant non-lipid metabolites limited the ability to conduct a similar enrichment analysis for non-lipids.

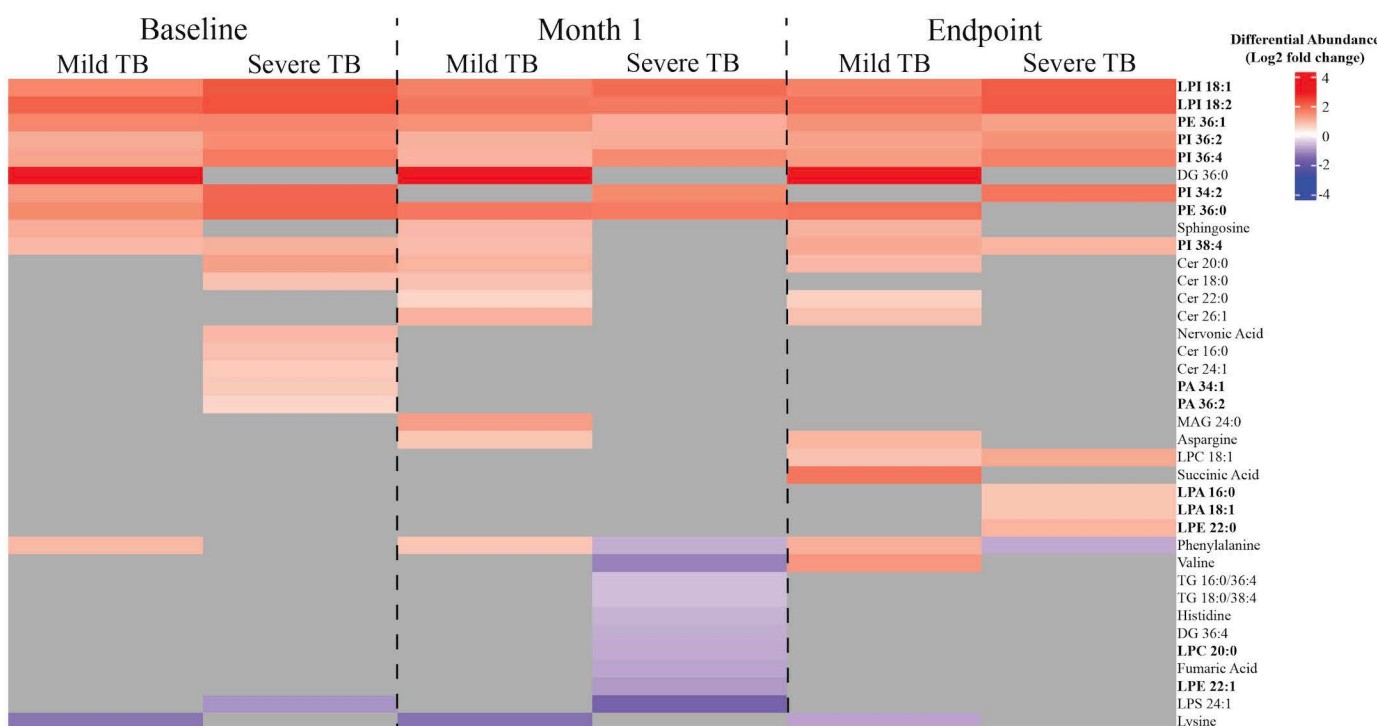

**Fig 1. Differentially abundant metabolites due to disease severity.** The log2 fold change differential abundance of metabolites influenced by disease severity in individuals with mild or severe TB at baseline (left), month 1 (middle), and month 6 (right) (p<0.05, Benjamini-Hochberg FDR correction, absolute log2 fold change>0.5). Metabolites that are not differentially abundant in each disease severity or treatment timepoint are shown in grey. Metabolite names highlighted in bold are glycerophospholipids.

PLOS Global Public
Health

## Differentially abundant disease severity-specific metabolites identified in glycerophospholipid and sphingolipid pathways

To identify disease severity-specific metabolites or metabolic pathways, we compiled the differentially abundant metabolites across all treatment timepoints in individuals with mild or severe TB. It was determined that while there were metabolites specific to each disease severity, some of the disease severity-specific metabolites contributed to the same metabolic pathways in individuals with mild or severe TB (Fig 3). For example, ceramide (Cer) 18:0, Cer 20:0, Cer 22:0, Cer 26:1, and sphingosine had increased abundance in individuals with mild TB relative to the healthy controls (Fig 3B), but Cer 16:0, Cer 18:0, Cer 20:0, Cer 24:0 and sphingosine had increased abundance in individuals with severe TB, relative to healthy controls. The same trend was present in the glycerophospholipid metabolic pathway between individuals with mild or severe TB.

## Distinct glycerophospholipid profiles found in healthy individuals compared to individuals with mild TB.

Next, we determined the impact of the interaction of severity of TB and treatment time on metabolite abundance (Fig 4). We first tried to isolate the effect of only treatment time (rather than the effect of both treatment time and disease severity) on metabolite abundance which resulted in no differentially abundant metabolites (S3 Table). However,

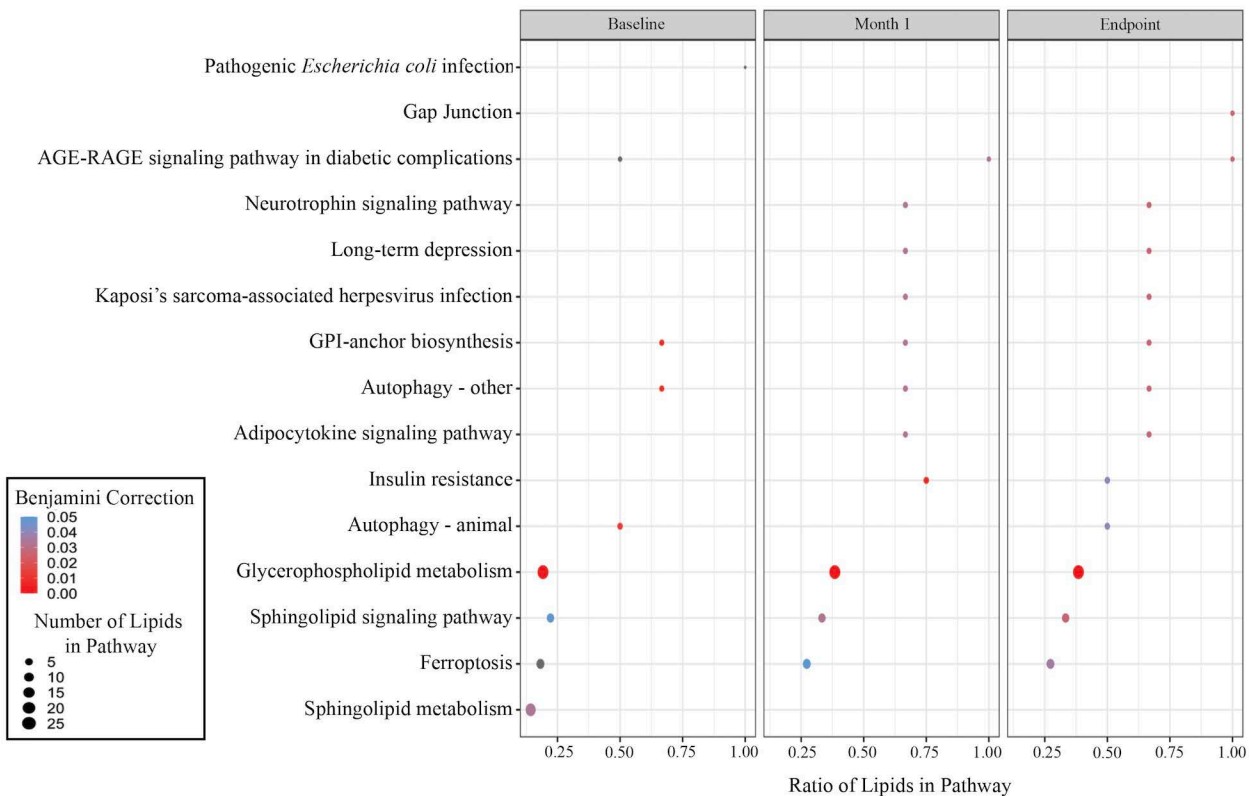

**Fig 2. Pathway enrichment of differentially abundant lipid metabolites due to disease severity.** The top 15 significantly enriched lipid metabolic pathways between individuals with mild and severe TB at baseline, month 1, and month 6. The color of the dots of a given pathway and treatment timepoint corresponds to the Benjamini-Hochberg FDR-corrected p-value (p-value < 0.05, absolute log2 fold change > 0.5). The size of the dots represents the total number of lipids from our analysis in a given metabolic pathway. The x-axis shows the ratio of lipids in a given pathway (number of differentially abundant lipids in a pathway/total number of lipids in the pathway).

A) Differentially abundant metabolites from individuals with mild TB compared to the controls across all timepoints

B) Differentially abundant metabolites from individuals with severe TB compared to the controls across all timepoints

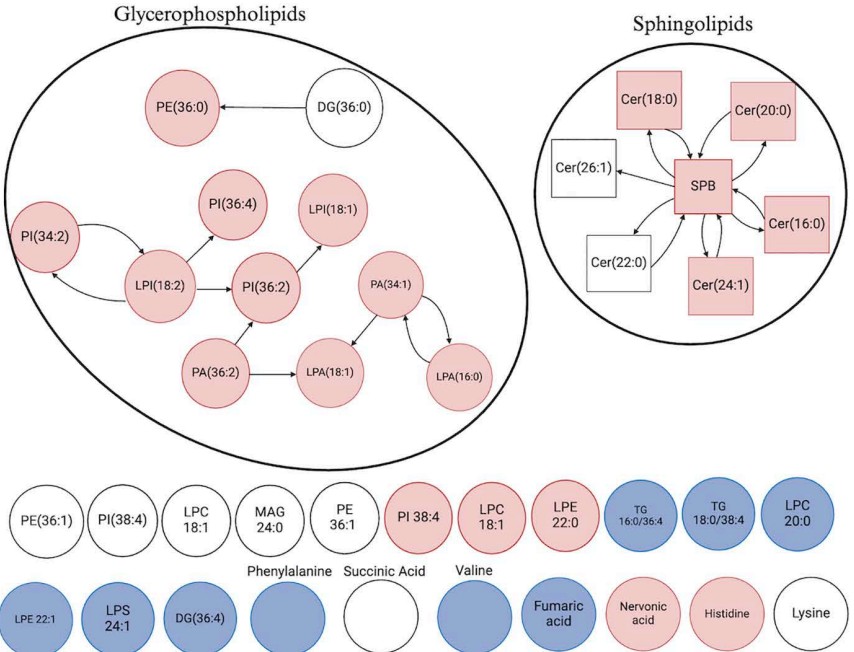

**Fig 3. Metabolite interaction network of the combined differentially abundant metabolites influenced by disease severity.** The interactions between differentially abundant metabolites influenced by disease severity in individuals with A) mild TB and B) severe TB compared to the controls ($p < 0.05$, Benjamini-Hochberg FDR correction, absolute log2 fold change $> 0.5$). While this analysis was conducted at each treatment timepoint, we combined the differentially abundant metabolites from each timepoint for these networks. Arrows indicate whether a given class is a precursor for another. If the metabolite is red, it signifies that it has increased abundance, blue signifies decreased abundance, and white signifies that it is not differentially abundant. Created in BioRender (2024).

PLOS Global Public Health

this model assumed that the relationship between treatment time and AUC of metabolites is the same across all groups. Therefore, the model testing the interaction of severity of TB and treatment time was likely more suitable for the data.

Our analysis testing the interaction between severity of TB and treatment time on metabolite abundance resulted in no differentially abundant metabolites in individuals with severe TB at any treatment time (Fig 4). Conversely, in individuals with mild TB, we found that phosphatidylcholine (PC) 36:1, cholesterol, L-lysine, monoacylglycerol (MAG) 18:1, and MAG 20:4 had decreased abundance compared to the controls (Fig 5), while there was an increased abundance of glycerophospholipids at baseline. These results support our results (Fig 1) where there was increased glycerophospholipid abundance individuals with mild TB compared to the controls using a separate model (Fig 1, Fig 5). Interestingly, the controls showed a decrease in glycerophospholipid abundance while individuals with mild TB showed an increase at baseline (Fig 5). These findings show distinct glycerophospholipid profiles between controls and individuals with TB.

## Discussion

To our knowledge, this is the first study to longitudinally compare the metabolomic profiles of individuals with pulmonary TB stratified by both disease severity (based on CXR and smear grade) and treatment duration over the full six-month course of therapy. While there were differentially abundant metabolites specific to individuals with mild compared to

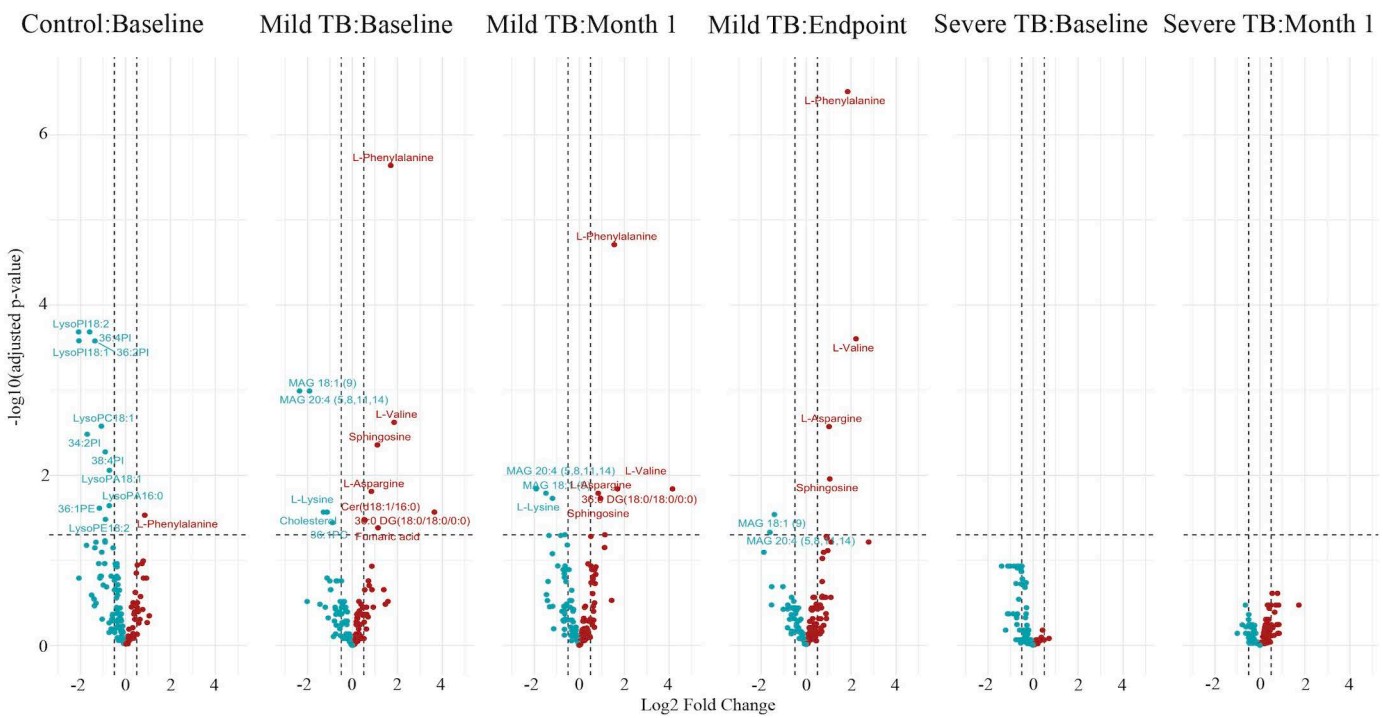

**Fig 4. Differentially abundant metabolites influenced by both disease severity and treatment time.** A volcano plot of the differentially abundant metabolites influenced by the interaction between treatment time and disease severity. The blue dots represent metabolites with decreased abundance as compared to the control while the red dots represent increased abundance as compared to the control. The x-axis shows the log2 fold change while the y-axis represents the –log10 adjusted p-value. The dotted line on the x-axis shows the log2 fold change cut off of 0.5 that was used in our analysis while the y-axis dotted line represents the p-value cut off of 0.05.

severe TB, the metabolites specific to either disease severity were involved in the glycerophospholipid and sphingolipid metabolic pathways (Fig 3). These pathways have known involvement in Mtb reproduction and virulence [5,42–48]. Glycerophospholipids, essential components of cell membranes, play crucial roles in cellular structure and signaling pathways [5,42,43]. The observed enrichment suggests a potential strategy employed by Mtb to modulate host lipid metabolism for its survival and proliferation Similarly, the enrichment of sphingolipids, which are involved in various cellular processes including cell signaling and apoptosis, highlights the complex interplay between Mtb and host lipid metabolism [44,45].

We hypothesized that TB infection would lead to elevated glycerophospholipid levels, which would stabilize by the end of anti-TB treatment, as the most intensive phase of therapy and immune response activity would have peaked by the treatment midpoint. Interestingly, glycerophospholipid abundance was increased at all timepoints in individuals with mild or severe TB as compared to the healthy controls. These results suggest that despite anti-TB treatment, Mtb could be chronically influencing glycerophospholipid profiles of Mtb-infected individuals. In previous research, Chen et al. 2021 found similar results in which glycerophopholipids were enriched in Chinese individuals with TB (N = 30) compared to controls (N = 30) over the course of anti-TB treatment [20].

Comparing our enrichment analysis to that of Chen et al. (2021) revealed consistent findings, with the highest enrichment at specific timepoints observed in both studies. These included pathways such as sphingolipid signaling, sphingolipid metabolism, glycosylphosphatidylinositol (GPI)-anchor biosynthesis, glycerophospholipid metabolism, and autophagy [20]. The small sample size in our study reflects strict inclusion and exclusion criteria aimed at ensuring well-defined, homogenous

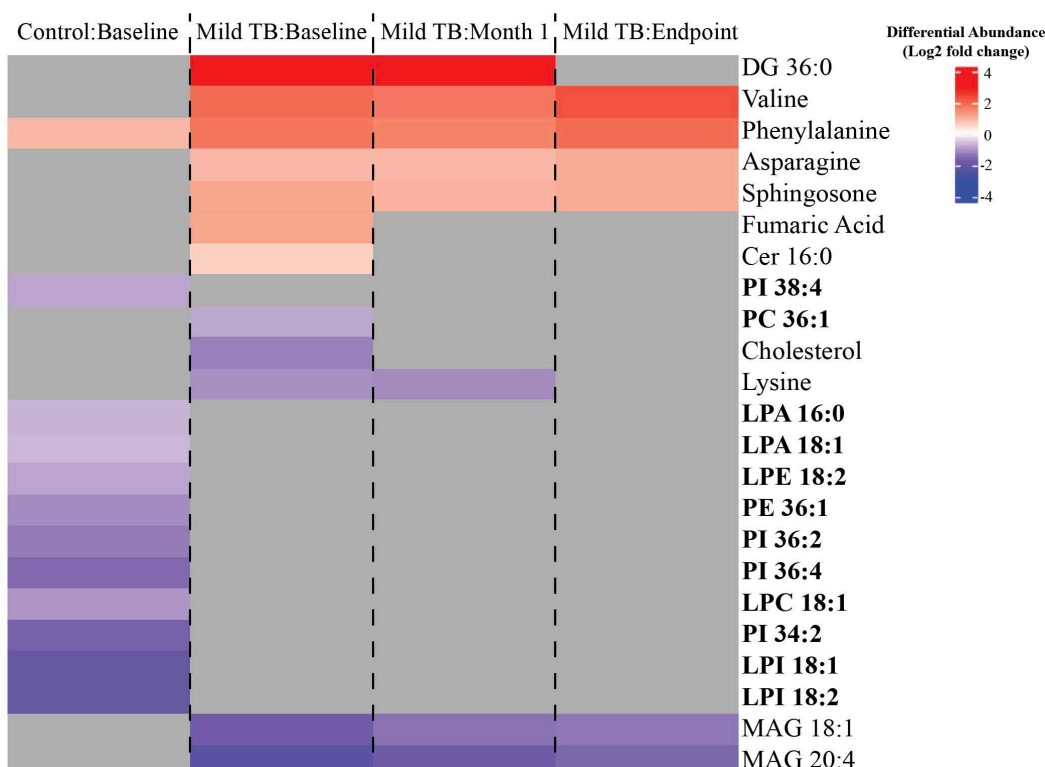

**Fig 5. Heat map of differentially abundant metabolites influenced by both disease severity and treatment time in individuals with mild TB.** The log2 fold change differential abundance of metabolites influenced by disease severity and treatment time in individuals with mild TB as compared to the controls (p < 0.05, Benjamini-Hochberg FDR correction, absolute log2 fold change > 0.5). Metabolites that are not differentially abundant in a given disease severity or treatment timepoint are shown in grey. Metabolite names highlighted in bold are glycerophospholipids.

groups and minimizing confounders (e.g., diabetes, HIV). Due to the retrospective nature of sample availability, few participants met these stringent criteria. Nevertheless, our carefully selected cohort enables meaningful exploratory analysis. Notably, the consistency of our enrichment results with those from a larger cohort underscores the robustness of our findings.

Unlike the model testing disease severity across all timepoints, the model evaluating the interaction between disease severity and treatment phase identified differentially abundant metabolites exclusively in individuals with mild TB (Fig 4, Fig 5). These included PC 36:1, cholesterol, L-lysine, MAG 18:1, and MAG 20:4, all of which showed reduced abundance compared to healthy controls. Cholesterol, a key carbon source for Mtb, has been shown to promote infection, and lowering its levels can suppress Mtb [49]. L-lysine supports Mtb's cell wall structure, energy metabolism, and virulence [50,51]. Both cholesterol and L-lysine, along with PI 36:1, were reduced during the intensive treatment phase (baseline and month 1) but returned to control levels by month 6, suggesting their depletion was treatment-driven (Fig 5). Mtb also hydrolyzes monoacylglycerols (MAGs) to obtain nutrients and adapt to the host environment [43], which may explain their consistently decreased abundance across all treatment timepoints (Fig 4, Fig 5).

Additionally, this model revealed lower baseline glycerophospholipid abundance in healthy controls, while individuals with TB exhibited elevated levels, highlighting distinct metabolic profiles. If validated in larger datasets, these glycerophospholipid differences could serve as promising biomarkers for Mtb infection.

Possible heterogeneity in response to anti-TB treatment could have contributed to the absence of differentially abundant metabolites in individuals with severe TB in the model assessing the interaction between TB severity and time (Fig 4). Variability in treatment response among individuals with TB is well-documented and may arise due to factors such as drug resistance, host immune status, and genetic variability [52–55]. Additional variability could have been induced by differences in the timing of sample collection between individuals after anti-TB treatment initiation or difference in baseline characteristics. However, the small sample size prevented us from testing differences between individuals with different baseline characteristics between groups. The lack of significant findings in this model underscored the importance of accounting for such heterogeneity in future studies. All the participants in the study were under standard TB treatment regimens and were cured at the end of six months of anti-TB treatment. However, factors such as age and other lifestyle characteristics may have also contributed to the observed heterogeneity. These factors should be accounted for in a larger cohort, as adding them to our current models would likely have led to overfitting given the small sample size. Increasing the sample size would enhance statistical power and enable the identification of subtle yet clinically relevant metabolic alterations associated with TB severity and treatment response.

We found that glycerophospholipids such as LysoPI 18:1, LysoPI 18:2, PE 36:1, and PI 36:4 were differentially abundant in individuals with TB regardless of disease severity and treatment timepoint. Furthermore, disease severity-specific metabolites involved in glycerophospholipid and sphingolipid metabolic pathways varied, suggesting that while disease severity affects metabolite abundance, these changes are linked to the same biological pathways, potentially influenced by anti-TB treatment. These findings highlight the importance of identifying metabolites that are consistently differentially abundant across both disease severities and all treatment timepoints, as such metabolites may serve as the most effective biomarkers for TB.

Our study underscores the need for larger cohort studies to validate these findings and explore additional factors influencing TB-associated metabolic alterations such as disease severity. These analyses provide novel insights into the metabolic alterations associated with TB severity at different treatment timepoints, highlighting the intricate interplay between Mtb infection and host metabolism.

## Supporting information

**S1 Table. Full list of Metabolites and lipids included in the study's Personal Compound Database Library.** The 261 surveyed metabolites and lipids sorted by their masses in ascending order. The ones that passed the stringency criteria of detection (165 lipids/metabolites) and that were used in this study are highlighted in bold.
(XLSX)

**S2 Table. The distribution of the demographic characteristics of Case and Controls.** A summary of the demographic characteristics of control, mild, and severe cases. Key variables include age, BMI, marital status, occupation, income, smoking, alcohol use, weight loss, and sputum conversion, highlighting differences in distributions across the groups. (JPG)

**S3 Table. The results of the treatment timepoint model.** We tested the effect of solely treatment time (rather than including both treatment time and disease severity) and found no significantly differentially abundant metabolites as shown in this excel document. No adjusted p values (adj.P.Val column) for any metabolite were below 0.05. (CSV)

## Acknowledgments

The authors would like to extend heartfelt gratitude to the RePORT India Consortium and each of the participants from the study sites for their kind participation.

## Author contributions

**Conceptualization:** Senbagavalli Prakash Babu, Komala Ezhumalai, Kalaivani Raghupathy, Vijay Viswanathan, Prakash Babu Narasimhan, Saurabh Mehta, Sonali Sarkar.

**Data curation:** Sonali Sarkar.

**Formal analysis:** Catherine H Kagemann.

**Funding acquisition:** Senbagavalli Prakash Babu.

**Investigation:** Senbagavalli Prakash Babu, Komala Ezhumalai, Arnab Chakraborty, Kalaivani Raghupathy, Siddesh S Kamat, Vijay Viswanathan.

**Methodology:** Senbagavalli Prakash Babu, Komala Ezhumalai, Arnab Chakraborty, Kalaivani Raghupathy, Siddesh S Kamat, Vijay Viswanathan, Prakash Babu Narasimhan, Saurabh Mehta, Sonali Sarkar.

**Project administration:** Senbagavalli Prakash Babu, Samantha L Huey, Saurabh Mehta, Sonali Sarkar.

**Supervision:** Senbagavalli Prakash Babu, Kalaivani Raghupathy, Saurabh Mehta.

**Visualization:** Catherine H Kagemann.

**Writing – original draft:** Catherine H Kagemann, Arnab Chakraborty, Komala Ezhumalai, Saurabh Mehta.

**Writing – review & editing:** Senbagavalli Prakash Babu, Komala Ezhumalai, Samantha L Huey, Pranay Sinha, Elaine A Yu, Saurabh Mehta.

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
