## [Decision Letter · Decision Letter 0]

7 May 2025

PGPH-D-25-00214

Metabolite dynamics over the course of anti-tuberculosis treatment in individuals with mild and severe tuberculosis

Dear Dr. Mehta,

Thank you for submitting your manuscript to PLOS Global Public Health. After careful consideration, we feel that it has merit but does not fully meet PLOS Global Public Health’s publication criteria as it currently stands. Therefore, we invite you to submit a revised version of the manuscript that addresses the points raised during the review process.

We look forward to receiving your revised manuscript.

Kind regards,

Kiatichai Faksri, Ph.D

Academic Editor

Journal Requirements:

Additional Editor Comments (if provided):

Dear Authors

Thank you for submitting your manuscript entitled "Metabolite dynamics over the course of anti-tuberculosis treatment in individuals with mild and severe tuberculosis." As you can see, the reviewers suggest for a minor revision according to the reviewer comments attached. Therefore, I suggest you to submit the revised version accordingly.

Your study presents a well-structured and insightful pilot investigation that explores disease-severity-specific metabolic profiles during TB treatment, including among control participants. The integration of metabolomics with clinical indicators such as chest X-ray (CXR) and sputum smear grading is commendable and adds significant value to the findings.

The manuscript is generally well-written, and your use of appropriate analytical methods, including computational and statistical approaches, strengthens the study. Nonetheless, there are a few areas that require minor revisions to improve clarity and completeness:

Introduction: Although concise, the introduction would benefit from a brief contextualization of the global and national burden of tuberculosis, as well as a short explanation of the importance of metabolomics in understanding TB pathogenesis and treatment response. This will enhance the rationale for your study.

Inclusion Criteria: Please clearly define the inclusion and exclusion criteria for study participants. For instance, were participants selected based on sex, age, or absence of comorbidities? Explicitly stating this will improve the reproducibility of your findings.

Results and Conclusion: The Results and Conclusion sections would benefit from additional explanation and clarity. Some of the findings are currently presented too briefly, which may hinder full understanding of their significance and implications.

In-Text Citations: Please review your in-text citations for accuracy. For example, the citation in line 118 currently refers to reference 27 but should be corrected to reference 2. A general review of all citation numbers is recommended to ensure consistency.

Sample Size Justification: While you acknowledge the limited sample size, a brief justification or contextual note regarding sample size estimation—even within the scope of a pilot study—would strengthen the manuscript’s methodological transparency.

Overall, this is a solid and promising proof-of-concept study. The suggested revisions are relatively minor and aimed at improving clarity and contextualization.

Sincerely yours

Prof. Kiatichai Faksri

Academic Editor

Reviewers' comments:

Reviewer's Responses to Questions

**Comments to the Author**

1. Does this manuscript meet PLOS Global Public Health’s publication criteria?

Reviewer #1: Yes

Reviewer #2: Partly

2. Has the statistical analysis been performed appropriately and rigorously?

Reviewer #1: Yes

Reviewer #2: Yes

3. Have the authors made all data underlying the findings in their manuscript fully available (please refer to the Data Availability Statement at the start of the manuscript PDF file)?

Reviewer #1: Yes

Reviewer #2: Yes

4. Is the manuscript presented in an intelligible fashion and written in standard English?

Reviewer #1: Yes

Reviewer #2: Yes

Reviewer #1: Kagemann et al. investigated metabolite dynamics over the course of anti-TB treatment in individuals diagnosed with mild and severe PTB, revealing disease-severity-specific metabolic profiles even in the controls. The manuscript is well-structured, with a clear introduction, aim, materials and methods (metabolomic and computational analyses, the right statistical analysis), results, and discussion. It is also commending that CXR and sputum smear grading methods were used.

This is a pilot proof-of-concept study, and it is a good one. However, there are a few areas that need clarification:

1. Introduction: Although the introduction is brief and concise, it will be informative to provide a global and contextual (country-wise) burden of TB. A brief importance of metabolomics study in the understading of TB treatment dynamics.

2. Inclusion Criteria: It would be beneficial to explicitly state the inclusion criteria for the study participants. For example, are they male subjects without comorbidities?

3. Results and Conclusion Sections: The results and conclusion sections of the study require more detailed explanations to ensure clarity and understanding.

4. In-Text Citations: The in-text citations need to be thoroughly reviewed. For instance, the citation in line 118 should be corrected from 27 to 23

Reviewer #2: This is a well-written and sound paper. However, one key omission is the calculated sample size. While the authors mention that the sample size is small, they do not provide context regarding the required or estimated sample size, making the statement ambiguous.

**Do you want your identity to be public for this peer review?** For information about this choice, including consent withdrawal, please see our Privacy Policy

Reviewer #1: No

Reviewer #2: No

---

## [Decision Letter · Decision Letter 1]

1 Jul 2025

Metabolite dynamics over the course of anti-tuberculosis treatment in individuals with mild and severe tuberculosis

PGPH-D-25-00214R1

Dear Dr. Mehta,

We are pleased to inform you that your manuscript 'Metabolite dynamics over the course of anti-tuberculosis treatment in individuals with mild and severe tuberculosis' has been provisionally accepted for publication in PLOS Global Public Health.

Best regards,

Kiatichai Faksri, Ph.D

Academic Editor

Dear Dr. Saurabh Mehta

On behalf of the PLOS Global Public Health Editorial Team, I am delighted to inform you that your manuscript entitled

“Metabolite dynamics over the course of anti-tuberculosis treatment in individuals with mild and severe tuberculosis” has been formally accepted for publication.

We congratulate you and your co-authors on this achievement.

Thank you for choosing PLOS Global Public Health for your work. We look forward to publishing your valuable contribution to the field of tuberculosis research.

Should you have any questions during the production process, please do not hesitate to contact us at pgph-production@plos.org.

Once again, congratulations on your successful acceptance. We wish you every success with your forthcoming publication.

Sincerely,

Kiatichai Faksri

Academic Editor

Reviewer Comments (if any, and for reference):

Reviewer's Responses to Questions

**Comments to the Author**

Reviewer #1: All comments have been addressed

Reviewer #2: All comments have been addressed

publication criteria?

Reviewer #1: Yes

Reviewer #2: Yes

3. Has the statistical analysis been performed appropriately and rigorously?

Reviewer #1: Yes

Reviewer #2: Yes

4. Have the authors made all data underlying the findings in their manuscript fully available (please refer to the Data Availability Statement at the start of the manuscript PDF file)?

Reviewer #1: Yes

Reviewer #2: Yes

5. Is the manuscript presented in an intelligible fashion and written in standard English?

Reviewer #1: Yes

Reviewer #2: Yes

Reviewer #1: The issues raised in the earlier version of the manuscript have been thoroughly addressed and submitted as the revised version. I recommend this manuscript for publication in your highly read journal.

Reviewer #2: All the comments were addressed satisfactorily

**Do you want your identity to be public for this peer review?** For information about this choice, including consent withdrawal, please see our Privacy Policy

Reviewer #1: **Yes: ** Samuel Ayanwale

Reviewer #2: No
